# Bidirectional contact tracing could dramatically improve COVID-19 control

William J. Bradshaw[1,2], Ethan C. Alley[2,3], Jonathan H. Huggins[4], Alun L. Lloyd [5] & Kevin M. Esvelt [3✉]

Contact tracing is critical to controlling COVID-19, but most protocols only "forward-trace" to notify people who were recently exposed. Using a stochastic branching-process model, we find that "bidirectional" tracing to identify infector individuals and their other infectees robustly improves outbreak control. In our model, bidirectional tracing more than doubles the reduction in effective reproduction number ($R_{eff}$) achieved by forward-tracing alone, while dramatically increasing resilience to low case ascertainment and test sensitivity. The greatest gains are realised by expanding the manual tracing window from 2 to 6 days pre-symptom-onset or, alternatively, by implementing high-uptake smartphone-based exposure notification; however, to achieve the performance of the former approach, the latter requires nearly all smartphones to detect exposure events. With or without exposure notification, our results suggest that implementing bidirectional tracing could dramatically improve COVID-19 control.

[1] Max Planck Institute for Biology of Ageing, Joseph-Stelzmann-Str. 296, 50937 Cologne, Germany. [2] Alt. Technology Labs, Berkeley, CA 94702, USA. [3] Media Laboratory, Massachusetts Institute of Technology, Cambridge, MA 02139, USA. [4] Department of Mathematics & Statistics, Boston University, Boston, MA 02215, USA. [5] Biomathematics Graduate Program and Department of Mathematics, North Carolina State University, Raleigh, NC 27695, USA. ✉email: esvelt@mit.edu

Contact tracing, isolation, and testing are some of the most powerful public health interventions available. The nations that have most effectively controlled the ongoing COVID-19 pandemic are noteworthy for conducting comprehensive and sophisticated tracing and testing[1]. Current "forward-tracing" protocols seek to identify and isolate individuals who may have been infected by the known case, preventing continued transmission (Fig. 1a). For example, the European Union and World Health Organization call for the identification of potential infectees starting 2 days prior to the development of symptoms[2,3].

However, chains of SARS-CoV-2 transmission may persist despite excellent medical monitoring and forward-tracing programs due to substantial rates of undiagnosed or asymptomatic transmission[4] (Fig. 1a). Asymptomatic carriers, who reportedly bear equivalent viral loads to patients exhibiting symptoms[5], have been estimated to account for 18%[6] to 79%[7] of cases, with multiple population surveys indicating intermediate values around 45%[8–10]. Moreover, a large fraction of transmission is known to be driven by superspreading events[11], suggesting that methods that preferentially identify, test and trace potential superspreaders would be especially valuable in reducing transmission.

We hypothesized that "bidirectional" contact tracing could identify and isolate undiscovered branches of the transmission tree, preventing many additional cases—especially when asymptomatic carriers are common or case ascertainment rates are low (Fig. 1b). Bidirectional contact tracing uses "reverse-tracing" to identify the parent case who infected a known case, then continues tracing to iteratively discover other cases related to the parent. It has been successfully used to identify clusters and community transmission in Japan[12] and Singapore[13,14], but is otherwise uncommon. Previous studies of COVID-19 contact tracing have largely neglected the possibility of gains from bidirectional tracing, with most models designed such that only forward-tracing can occur.

We further hypothesized that bidirectional tracing would be most effective using a "hybrid" system that supplements manual tracing (Fig. 1c) with digital exposure notification. Numerous ongoing efforts aim to use smartphones emitting randomized Bluetooth and/or ultrasound "chirps" to notify people exposed to infected individuals (Fig. 1d, e). Digital approaches theoretically offer considerable advantages in speed[15], scale, efficacy[16], and confidentiality[17], suggesting that they may offer an effective method of implementing bidirectional tracing. However, their use in this context has not previously been investigated, and existing implementations primarily focus on "forward-notifying" cases exposed during the peak infectious window of the notifier[18].

To investigate the efficacy of bidirectional contact tracing and digital exposure notification, we adapted and extended a stochastic branching-process model of SARS-CoV-2 forward-tracing[19] and used it to explore the efficacy of different tracing strategies under plausible epidemiological scenarios. We find that either expanding the manual tracing window to enable more

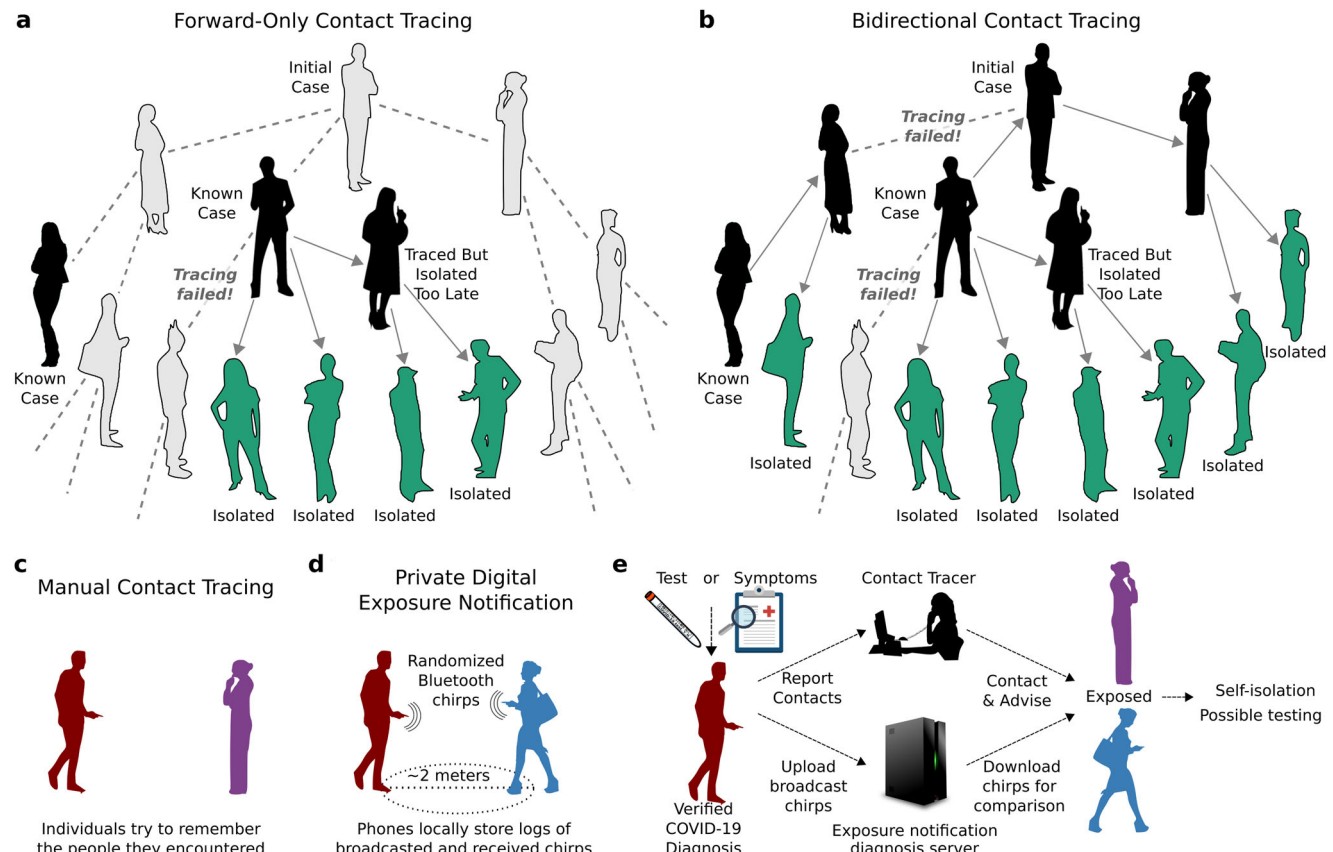

**Fig. 1 Forward-only and bidirectional contact tracing and digital exposure notification. a** Notifying people exposed to known cases (black) and isolating them (green) can prevent further transmission, but will miss asymptomatic and undiagnosed cases (gray) and descendants. **b** Bidirectional tracing also notifies and tests potential infectors, enabling isolation of additional cases. **c** Manual contact tracing requires individuals to share recent contacts with health authorities. **d** In digital exposure notification, smartphones broadcast rotating pseudorandom "chirps" and record those emitted by nearby devices[48]. **e** Individuals diagnosed with COVID-19 can "opt-in" by uploading broadcasted chirps to a diagnosis server[48]. All devices frequently check the server and alert the user if the calculated exposure exceeds a threshold set by the local health authority. In hybrid manual+digital systems, human tracers would seek to identify contacts without smartphones.

**Table 1 Key parameters of the branching-process model.**

| Parameter | Value | | | Sources and Notes |
|---|---|---|---|---|
| | Median | Optimistic | Pessimistic | |
| % asymptomatic carriers | 45% | 40% | 55% | 8-10,33 |
| Relative infectiousness of asymptomatic carriers | 50% | 45% | 60% | Informed by viral loads and tracing results described in refs. 5,9,26,34-36 |
| % environmental transmission | 10% | 5% | 15% | 16,37 |
| Proportion of pre-symptomatic transmission | 48% | 38% | 53% | Informed by refs. 5,26,36,38-43 |
| % of cases with chirping smartphones | 53% (low uptake)/80% (high uptake) | | | Survey data22-24 |
| % of symptomatic cases identified without tracing | 50% | | | 4 |
| Test sensitivity | 70% | | | 44,45 |
| $R_0$ (default) | 2.5 | | | Most estimates cluster between 2.0 and 3.0:refs. 9,16,26-28,46 |
| Incubation period | 5.5±2.1 days (lognormal distribution) | | | 20 |
| Delay from onset to isolation | 3.8±2.4 days (Weibull distribution) | | | 19 |
| Delay for testing | 1±0.3 days (gamma distribution) | | | Assumed |
| Delay for manual tracing | 1.5±4.8 days (lognormal distribution); median 0.5 days | | | Previous reports suggest most contacts can be traced within one day, but some take much longer47 |
| Delay for digital tracing | 0 days | | | Assumed |

effective bidirectional tracing or implementing high-uptake bidirectional digital exposure notification could substantially improve COVID-19 control.

## Results

In our model, each case generates a number of new cases drawn from a negative binomial distribution, with incubation and generation-time distributions based on the published literature (Table 1 and Supplementary Table 1). Cases could be identified and isolated based on symptoms alone or through contact tracing (Methods). We assumed that symptomatic cases required a positive test before initiating contact tracing, as is the case in the EU[2] and most US jurisdictions. Ninety percent of cases were assumed to comply with isolation, after which they generated no further child cases. Each outbreak was initialized with 20 index cases to minimize stochastic extinction and designated as "controlled" if it reached extinction (zero new cases) before reaching 10,000 cumulative cases. Effective reproduction numbers ($R_{eff}$) were computed as the mean number of child cases produced per case.

We began by investigating a median scenario in which 10% of transmission was assumed to be environmental (and therefore untraceable), 48% of transmission occurred pre-symptomatically, and 45% of cases were asymptomatic with 50% infectiousness. For the initial analysis we assumed a fixed basic reproduction number ($R_0$) of 2.5, a 50% ascertainment rate for symptomatic cases, and a test sensitivity of 70%, but explored other values below.

**Bidirectional manual tracing with an expanded tracing window could more than double efficacy.** In our initial scenario, manual forward tracing and isolation of contacts occurring up to 48-h before symptom onset or diagnosis per current guidelines[2] is predicted to reduce $R_{eff}$ by as much as 0.24 relative to the no-tracing baseline (Fig. 2a). Extending the time window for forward-tracing contacts beyond 2 days yielded negligible additional benefit (Fig. 2a and Supplementary Fig. 1). Switching from forward-only to bidirectional tracing without altering the 48-h tracing window further reduced $R_{eff}$ by up to 0.24, roughly doubling the benefit relative to no tracing.

Due to the extended generation and incubation times of COVID-19, the contact between an infector individual and their infectee will often occur more than 48-h prior to the latter's onset of symptoms[20]. Hence, extending the tracing window should substantially improve the efficacy of bidirectional tracing. As expected, extending the window for manual bidirectional tracing to 6 days pre-symptom onset resulted in a dramatic further reduction in $R_{eff}$ in our model, yielding values up to 0.42 lower than with a 2-day window (Fig. 2a, b and Supplementary Fig. 1)— an improvement in performance of roughly 85% relative to 2-day bidirectional tracing and 275% relative to forward-only tracing. Since real tracing programs—mostly focused on forward tracing with a 2-day window[2,3]—are likely to detect some but not all reverse contacts occurring up to 2 days before symptom onset, our model suggests the real gains of extending the tracing window would fall somewhere between these two extremes.

Our results demonstrate the importance of including bidirectional approaches when investigating the effectiveness of contact tracing. If they are borne out in practice, extending the tracing window to 6 days while swiftly testing early contacts would dramatically improve COVID-19 control.

**Digital exposure notification is fragile to network fragmentation.** In principle, digital exposure notification can instantaneously notify all contacts recorded by a mobile device for the

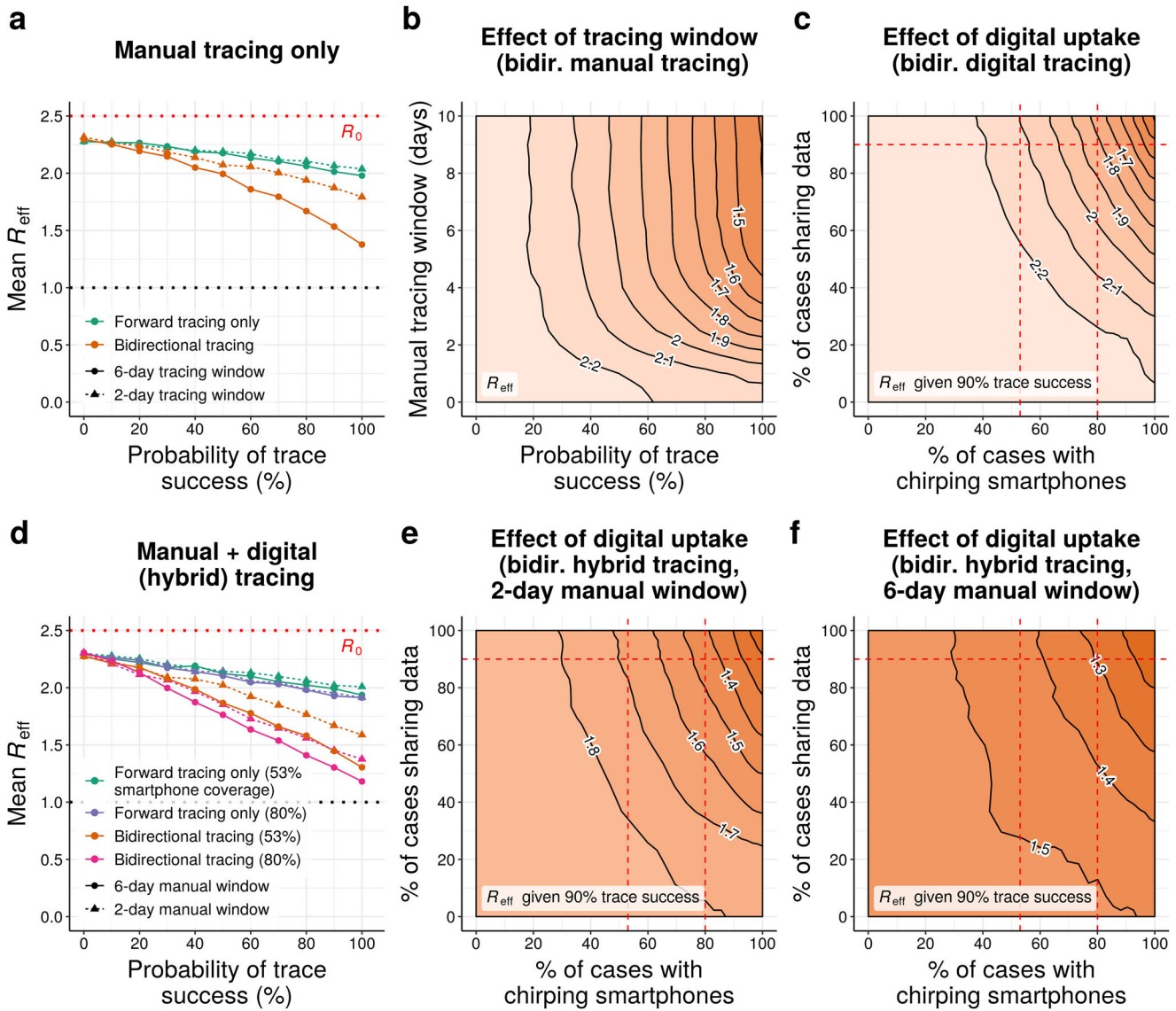

**Fig. 2 Comparing forward and bidirectional contact tracing at $R_0 = 2.5$. a** Mean $R_{eff}$ achieved by manual tracing with 2-day and 6-day manual tracing windows. **b** Neighbor-averaged contour plot, showing mean $R_{eff}$ achieved by bidirectional manual tracing as a function of the probability of trace success and the width of the manual tracing window. **c** Neighbor-averaged contour plot, depicting $R_{eff}$ achieved by digital exposure notification in the absence of manual tracing as a function of smartphone coverage and data-sharing, assuming 90% probability of trace success. **d** Mean $R_{eff}$ achieved using hybrid manual+digital tracing, assuming 90% data sharing and 53% or 80% smartphone coverage. **e, f** Neighbor-averaged contour plots, depicting $R_{eff}$ achieved by hybrid tracing as a function of smartphone coverage and data-sharing, assuming 90% probability of trace success and a **e** 2-day or **f** 6-day manual tracing window. All panels assume median disease parameters (Table 1). "Probability of trace success" refers to trace attempts that are not otherwise blocked by environmental transmission or fragmentation of the digital network.

duration of the data-retention period, which is typically 14 days[21]. Past studies of COVID-19 contact tracing have suggested that such methods could effectively control the pandemic, even in the absence of manual tracing programs[16]. In principle, the long data-retention period of digital tracing systems should enable a high degree of bidirectional tracing; however, previous studies have largely neglected this possibility, and existing digital tracing systems typically prioritize notifying contacts infected during peak infectiousness of the known case (i.e., forward tracing)[18].

In our model, implementing digital exposure notification in the absence of manual tracing was highly effective when all cases participate in the digital system, with $R_{eff}$ values approaching 1.0 (Fig. 2c and Supplementary Fig. 2). This performance, however, was highly sensitive to uptake of the digital system: even small decreases in the proportion of individuals carrying a participating

smartphone or (to a lesser extent) sharing their exposure data upon diagnosis resulted in a substantial increase in the effective reproduction number of the epidemic (Fig. 2c and Supplementary Fig. 3). Even if every individual with a smartphone (~80% of adults in the US[22]) participated in the system, and 90% of those shared their data upon diagnosis, the $R_{eff}$ achieved was only slightly lower than that of manual bidirectional tracing with a 2-day window, and substantially higher than manual-only tracing with a 6-day window (Fig. 2a, c). As a result of this fragility, our results suggest that digital exposure notification alone is unlikely to be a viable method for controlling COVID-19.

**Hybridizing manual tracing with digital may offer an alternative path to high performance.** In practice, almost no jurisdiction is proposing to exclusively control COVID-19 through

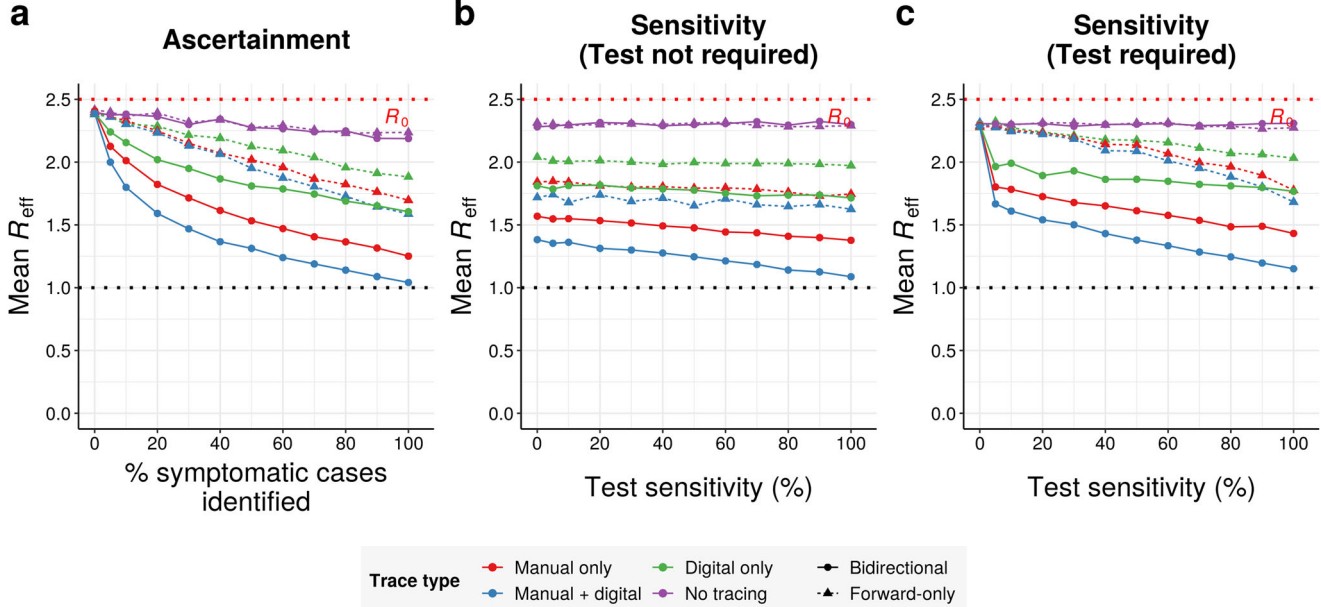

**Fig. 3 Bidirectional tracing under reduced ascertainment and test sensitivity. a** Mean $R_{eff}$ achieved by different tracing strategies as a function of the percentage of symptomatic cases that can be identified by health authorities on the basis of symptoms alone, assuming 70% test sensitivity. **b** Mean $R_{eff}$ achieved by different tracing strategies as a function of test sensitivity, assuming 50% ascertainment of symptomatic cases. **c** As in (**b**), but allowing contact tracing to be initiated from symptomatic cases on the basis of symptoms alone (i.e., without a positive test result). All panels assume 90% probability of trace success, a 6-day manual tracing window, high (80%) smartphone coverage, and median disease parameters (Table 1). Isolation on the basis of symptoms can dampen the outbreak even in the absence of tracing.

digital exposure notification, but rather to supplement traditional manual tracing with digital tools. The two methods have complementary strengths and weaknesses: digital tracing is fast, scalable, and could be easily adapted to trace bidirectionally, but is highly fragile to network fragmentation; manual tracing is slower and more labor-intensive, but more robust. A hybrid of the two approaches might thus outperform either approach used in isolation.

We investigated the effects of supplementing manual tracing with digital exposure notification for two distinct digital scenarios. In our "low-uptake" condition, 53% of cases possessed chirping smartphones, corresponding to roughly two-thirds of smartphone users; this is consistent with early survey data on willingness to download a contact-tracing app[23,24]. In our "high-uptake" condition, 80% of cases possessed chirping smartphones, corresponding to virtually all smartphone users in the US[22]. In both cases, we assumed that 90% of diagnosed cases upload their broadcasted chirps upon diagnosis.

In the absence of bidirectional tracing, a hybrid approach offered few benefits over manual tracing, reducing $R_{eff}$ by up to 0.06 in the low-uptake condition and 0.12 in the high-uptake condition compared to manual tracing alone (Fig. 2d and Supplementary Fig. 4). Bidirectional hybrid tracing with a 6-day manual window provided larger benefits compared to manual tracing, with a relative drop in $R_{eff}$ of up to 0.14 in the low-uptake condition and up to 0.26 in the high-uptake condition. By far the largest relative gains, however, were seen when the manual tracing window was restricted to 2 days pre-symptom onset: in this case, supplementing manual with digital tracing reduced $R_{eff}$ by up to 0.21 in the low-uptake case and up to 0.42 in the high-uptake case compared to manual tracing alone. Since manual bidirectional tracing with a 2-day window achieved a reduction in $R_{eff}$ of up to 0.48, these results suggest that implementing high-uptake exposure notification alongside manual tracing could roughly double the efficacy of current contact tracing efforts at reducing COVID-19 transmission.

When interpreting these results, it is important to distinguish between the absolute and relative efficacy of hybrid tracing. Increasing the manual tracing window from 2 to 6 days substantially increased the *absolute* efficacy of bidirectional hybrid tracing in both the high- and low-uptake conditions (Fig. 2d–f and Supplementary Fig. 5); however, since increasing the tracing window also increased the efficacy of manual-only bidirectional tracing (Fig. 2a), the *relative* benefit of supplementing manual with digital tracing was reduced (Supplementary Fig. 4). When the manual window is short, even low-uptake digital tracing can improve performance by providing an alternative avenue for effective bidirectional tracing; when it is long, only very high-uptake digital tracing offers a substantial additional advantage.

**Bidirectional tracing is more robust to low case ascertainment and test sensitivity.** Our results thus far assume that 50% of symptomatic cases can be identified based on symptoms alone, corresponding to an overall ascertainment rate (including asymptomatic cases) of roughly 22%. However, estimates of real-world symptomatic ascertainment rates for COVID-19 have varied dramatically country-by-country and over time, from under 10% in some of the worst-hit countries to over 90% in Australia[4]. To investigate the effect of case ascertainment on epidemic control, we varied the proportion of symptomatic cases identified, while holding the baseline probability of trace success (excluding environmental transmission and network fragmentation) constant at 90%.

Unsurprisingly, reducing ascertainment of symptomatic cases impaired epidemic control (Fig. 3a and Supplementary Figs. 6 and 27). However, bidirectional tracing was considerably more robust to poor case ascertainment than forward-only tracing, resulting in dramatically lower $R_{eff}$ values across a wide range of ascertainment rates (Supplementary Fig. 7). When ascertainment rates are extremely low, as occurred in the United Kingdom

through much of the spring[4], bidirectional tracing dramatically outperforms current protocols.

We also investigated the effect of testing requirements and sensitivity on the efficacy of contact tracing. If symptomatic cases were traced immediately based on symptoms alone (without requiring a positive test result), both forward-only and bidirectional tracing were fairly robust to drops in test sensitivity, with $R_{eff}$ values increasing only slowly as sensitivity falls (Fig. 3b). Our baseline assumption, however, was that a positive test result was required to trace the contacts of any case, in line with common practice in the UK, US and EU[2]. Under these conditions, the efficacy of forward-tracing was dramatically more dependent on a high test sensitivity: low sensitivities yielded greatly increased $R_{eff}$ values in forward-only scenarios (Fig. 3c and Supplementary Figs. 6, 7, and 26), consistent with previous modeling studies reporting impaired performance under these conditions[25]. In sharp contrast, the performance of bidirectional tracing remained relatively robust to changes in test sensitivity.

**The predicted benefits of bidirectional tracing are robust to changing the basic reproduction number.** To evaluate the epidemiological robustness of our findings, we repeated our analysis using $R_0$ values ranging from 1.0 to 4.0 (Fig. 4, left)[9,16,26–28]. We assumed a 90% baseline trace success probability, a 6-day manual trace window, immediate tracing of symptomatic cases, and high uptake of the digital system when present. A wider range of assumptions are explored in Supplementary Figs. 8–28.

For all tracing strategies investigated, $R_{eff}$ varied roughly linearly as a function of $R_0$ (Fig. 4, top-left). Bidirectional hybrid tracing was consistently the most effective strategy, with $R_{eff}$ values roughly 85% of those achieved under bidirectional manual tracing (Supplementary Fig. 29). The exception to this was when

$R_{eff} < 1$, in which case the $R_{eff}$ values achieved by different strategies converged (Fig. 4 and Supplementary Fig. 29). Both manual and hybrid bidirectional tracing dramatically outperformed forward-only approaches across a wide range of $R_0$ values. Reducing uptake of the digital system from 80% to 53% of cases largely abrogated the advantage of hybrid over manual approaches (Supplementary Fig. 9). However, if the manual tracing window was also constrained to 2 days pre-symptom onset per current protocols, even low-uptake hybrid tracing substantially outperformed the manual approaches (Supplementary Figs. 10 and 11).

Unlike $R_{eff}$, the proportion of outbreaks controlled was highly nonlinear with varying $R_0$ (Fig. 4, bottom-left). Within a critical window, small reductions in $R_0$ resulted in large increases in control probability across all forms of tracing. Hybrid bidirectional tracing exhibited the greatest degree of outperformance when $1.25 < R_0 < 3.25$. When $R_0 \leq 1.25$, manual and hybrid tracing both achieved nearly 100% control, while when $R_0 \geq 3.25$, no strategy achieved control probabilities over 10%.

**The predicted benefits of bidirectional tracing are robust to changing transmission parameters.** While COVID-19 is clearly a challenging disease to control, there remains substantial uncertainty around the exact rates of asymptomatic, presymptomatic, and environmental transmission. To explore a wider range of scenarios, we aggregated our collective best estimates to define optimistic and pessimistic values for these parameters, with 5/15% environmental transmission, 38/53% pre-symptomatic transmission, and 40/55% asymptomatic carriers, which were 45/60% as infectious as symptomatic cases. We repeated our simulations under these new assumptions for a range of $R_0$ values (Fig. 4, middle and right).

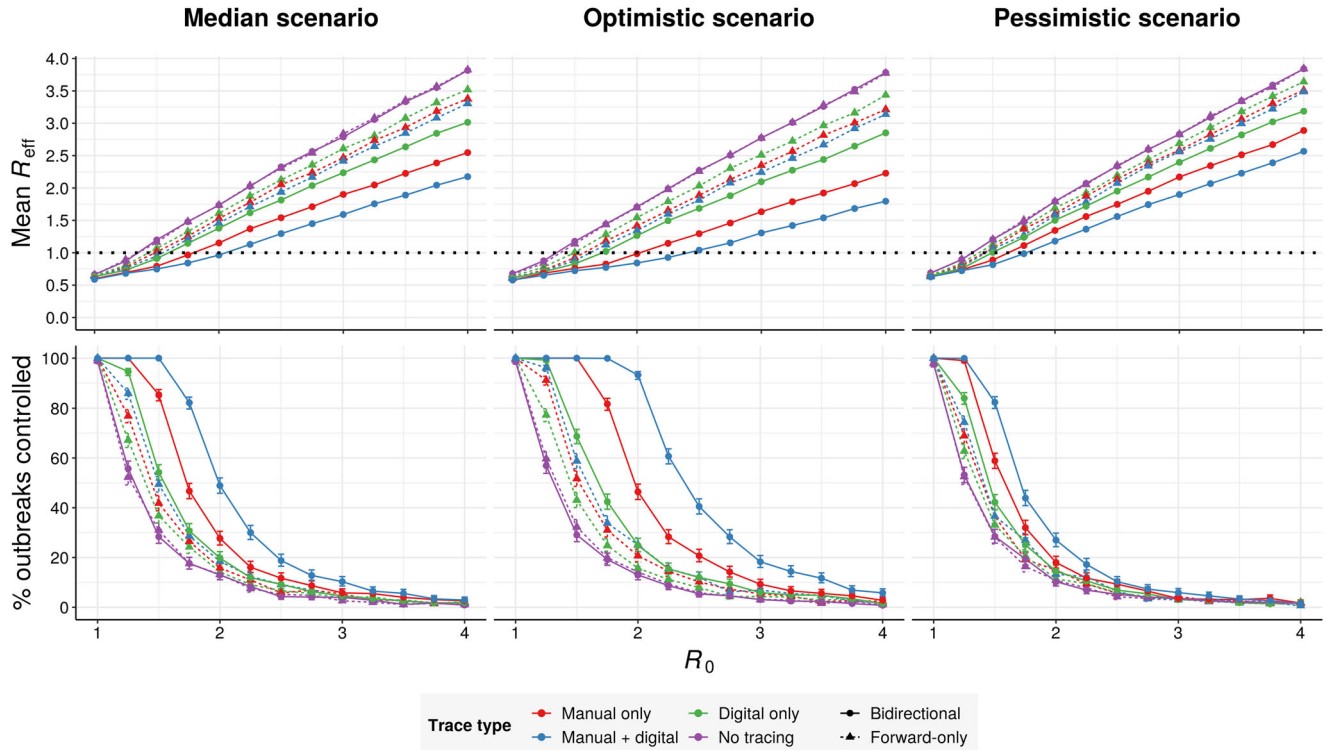

**Fig. 4 Effect of $R_0$ and disease parameters on performance.** (top row) Mean $R_{eff}$ achieved and (bottom row) mean % of outbreaks controlled by different tracing strategies as a function of the basic reproduction number $R_0$, assuming (left) median, (middle) optimistic or (right) pessimistic disease parameters (Table 1), assuming 50% ascertainment of symptomatic cases, 70% test sensitivity, 90% probability of trace success, a 6-day manual tracing window, and high (80%) smartphone coverage (Table 1). Error bars in the bottom row represent 95% credible intervals across 1000 runs under a uniform beta prior. Isolation on the basis of symptoms can dampen the outbreak at low $R_O$, even in the absence of tracing.

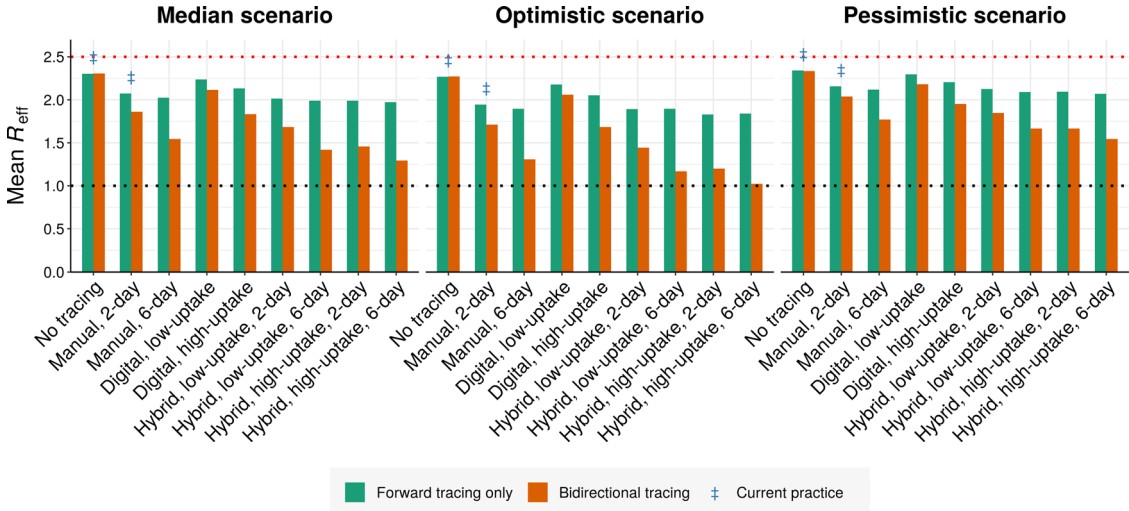

**Fig. 5 Performance of different tracing strategies relative to current practice.** Mean effective reproduction number obtained under (left) median, (middle) optimistic, and (right) pessimistic scenarios (Table 1), assuming an $R_0$ of 2.5 and a 90% baseline probability of trace success across 1000 runs. Blue double dagger symbols indicate conditions roughly corresponding to current practice in most regions. Low and high uptake correspond to 53% and 80% of cases, respectively, having chirp-enabled smartphones. Without tracing, forward and bidirectional are equivalent.

While hybrid bidirectional tracing continued to robustly outperform other configurations (Supplementary Figs. 8–12) in terms of $R_{eff}$, the probability of outbreak control varied substantially between scenarios. In the optimistic scenario, high-uptake hybrid bidirectional tracing was sufficient to reliably control outbreaks whenever $R_0 \leq 1.75$, while in the pessimistic scenario reliable control was only achieved at $R_0 \leq 1.25$. Modifying the uptake of the digital system or the width of the manual tracing window had similar qualitative impacts in all scenarios (Supplementary Figs. 9–11).

To summarize the effects of different approaches, we compared the predicted $R_{eff}$ values achieved under all three scenarios in the absence of other interventions (Fig. 5 and Supplementary Figs. 30–32). Across scenarios, the $R_{eff}$ reduction achieved by 2-day bidirectional manual tracing was roughly 1.7–2× that achieved by forward-only tracing. Increasing the manual tracing window to 6-days further improved performance by roughly 75–95% relative to 2-day bidirectional tracing. Supplementing 2-day bidirectional manual tracing with low-uptake exposure notification improved the performance by 40–60%, while high-uptake exposure notification yielded an improvement of 90–105%.

## Discussion
Given the tremendous suffering inflicted by the COVID-19 pandemic and the critical role of contact-tracing systems in its control, there is an urgent need to optimize the implementation of these systems. To combat the high rates of asymptomatic carriers and low case ascertainment, tracing programs must use their limited resources to find as many new cases as possible.

Since 10–20% of infected individuals are responsible for most transmission events[11], individuals known to have infected at least one person are statistically much more likely to have infected others. This suggests that finding and tracing infectors may be important, especially when many cases are otherwise escaping detection. However, most existing tracing protocols and exposure notification systems ignore contacts from more than 48-h before symptom onset and consequently struggle to identify infectors. Even so, our model predicts that the benefit of tracing infectors for those few cases that develop rapidly enough is considerable: jurisdictions that swiftly test and trace all contacts, including the

few infectors among them, reduce the basic reproduction number of the virus by twice as much as those that only trace forwards.

Even the best 48-h manual tracing protocols could be substantially improved by extending the time window for identifying contacts. Our model suggests two possible ways to achieve this: firstly, by expanding the manual tracing window to detect and immediately test early contacts, and secondly, by implementing a comprehensive and high-uptake digital exposure notification system that can perform the same function. If the manual tracing window could be extended to 6 days pre-symptom onset while maintaining the proportion of contacts identified, or if almost all smartphone users participate in the digital exposure notification system, our model predicts a second approximate doubling of the $R_{eff}$ reduction achieved.

For different reasons, both of these conditions would be challenging to achieve in practice. Even accounting for recurring contacts, simply increasing the width of the manual tracing window would require many more contacts to be traced and quarantined per identified case, just when the greater effectiveness of bidirectional tracing would be finding more cases. The magnitude of this effect depends, among other things, on the network structure of the population and the number and frequency of recurring contacts—as an upper bound, assuming no recurring contacts and a uniform rate of new contacts per day, such a change would increase the number of contacts traced in proportion to the width of the tracing window. Our model, which takes no account of these factors, is ill-equipped to quantify this aspect of the policy question. Future work should investigate the efficacy of bidirectional tracing using a wider variety of epidemiological models, to better quantify the ratio of benefits versus costs.

While the magnitude of the cost of bidirectional tracing is difficult to quantify in the present setting, these costs can nevertheless be minimized through efficient prioritization of forward and backward tracing. Since people found through backward-tracing are unlikely to remain highly infectious, there is less of a case for quarantine in advance of a positive test. Therefore, manual programs might prioritize backward-tracing and testing contacts from 3–6 days before symptom onset, and forward-tracing from identified infectors. This focus on infection clusters would be similar to Japan's tracing program[12]. Exposure

notification could make a similar distinction by sending a different message to early contacts requesting that they get tested as soon as possible. Under this approach, the only increase in quarantine numbers would result from forward-tracing from confirmed infectors to identify their other potential infectees.

While the efficacy of bidirectional manual tracing depends on the ability to successfully implement an expanded tracing window with limited resources, digital exposure notification depends critically on the proportion of individuals participating in the system—in particular, the proportion of cases possessing a chirping smartphone. Doubling the effectiveness of the best current 48-h contact tracing programs would require a participation rate of 80% (i.e., almost every current smartphone user), a level unlikely to be achieved under the current plans for opt-in participation. Making all smartphones listen for and record chirps by default could weaken the dependence on the uptake, improving performance while avoiding the privacy implications of automatic chirping. As suggested above, prioritizing prompt testing and tracing of earlier contacts, while encouraging recent contacts to self-quarantine, could help dampen transmission while minimizing the number of false-positive notifications.

We stress-tested our conclusions with a wide range of plausible parameter combinations and possible values of $R_0$. Importantly, the relative efficacy of bidirectional tracing compared to forward-only further increases when symptomatic case ascertainment is below 50%, as has been true across much of the world. Since case ascertainment plummets when cases are surging, some regions have ceased contact tracing under such conditions. However, our results suggest that implementing some form of bidirectional tracing can maintain the utility of contact tracing even when a local epidemic appears out of control.

Despite this stress-testing, our conclusions must be considered in the context of our model, which, while less idealized than many of its predecessors, has limitations. In addition to only considering infected individuals, it makes no distinction between mild and severe symptoms and does not consider demographic, geospatial, or behavioral variation between cases. Since only true cases are included in the model, only the sensitivity of testing is considered; in reality, the balance between test sensitivity and specificity is a crucial trade-off. The efficacy of a real-world tracing program will also depend on the availability of timely COVID-19 testing[15] and high adherence to quarantine requests[16,19,29,30], while digital systems will require efficient algorithms with acceptable sensitivity and specificity.

These caveats aside, there is considerable evidence that bidirectional tracing can be feasibly implemented in practice. Locales such as Singapore[13,14] and Washington State[31] have employed bidirectional tracing to determine whether community transmission is occurring, while Japan's protocol explicitly aims to identify and trace individuals responsible for infection clusters[12]. Together with these promising empirical findings, our results indicate that bidirectional contact tracing could play an essential and potentially decisive role in controlling COVID-19 elsewhere in the world.

**Reporting Summary**. Further information on research design is available in the Nature Research Reporting Summary linked to this article.

## Data availability
The data supporting the findings of this study are in the main manuscript and the Supplementary Information, and are available at https://github.com/willbradshaw/covid-bidirectional-tracing.

## Code availability
Code for configuring and running the model is publicly available at https://github.com/willbradshaw/covid-bidirectional-tracing[32].

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

## Acknowledgements

We thank Aaron Bucher of the COVID-19 HPC Consortium and Amazon Web Services for granting us extra cloud compute credits. This work was supported by gifts from the Reid Hoffman Foundation and the Open Philanthropy Project (to K.M.E.) and cluster time granted by the COVID-19 HPC consortium (MCB20071 to K.M.E.). E.C.A. was supported by a fellowship from the Open Philanthropy Project. A.L.L. is supported by the Drexel Endowment (NC State University) and by the award CDC U01CK000587-01M001 from the US Centers for Disease Control and Prevention. The funders had no role in the research, writing, or decision to publish.

## Author contributions

K.M.E. conceived the study. J.H.H. and A.L.L. identified a suitable model framework. W.J.B. designed and programmed the adapted model, advised by the other authors. All authors independently estimated key model parameters based on the literature. W.J.B. ran all simulations and generated figures with assistance from E.C.A. All authors jointly wrote and edited the manuscript.

## Competing interests

The authors declare no competing interests.
