## [Peer Review File · Nature Communications]

Reviewers' Comments:

Reviewer #1:

Remarks to the Author:

I am happy with the answers of the authors. My only point remains that the costs involved with backward tracing (a larger time window) are not quantified.

In the answers to the reviewers, the authors state: "There is also no simple relationship between the number of cases discovered and the number of individuals tested/quarantined, as the latter depends on the network structure of the population, the number and frequency of recurring contacts, and the manner (e.g. behavioural vs physiological) in which superspreaders differ from the rest of the population. As a result, it is difficult to quantitatively assess the costs of bidirectional tracing without a model that explicitly distinguishes between contacts and cases."

I fully agree with this statement, and it would be good that this is mentioned in the main text as well, which only states: "Our model, which takes no account of uninfected individuals or repeated contacts, is ill equipped to quantify this aspect of the policy question."

I do not think it is essential for publication, but despite the fact that the model is ill-equipped, it may be possible to obtain some estimates of the costs associated with different time windows, e.g.,

- 1) what is the ratio of detected patients per time unit with the different strategies (this ignores the fact that many contacts who did not acquire the disease got screened).

- 2) Each individual has the same very high number of contacts per unit of time and there is random mixing in the population. Then the number of patients screened should be proportional to the number of patients for which the contacts are screened multiplies by the length of the time window.

My guess is that the true effort will be somewhere between these two cases, but having an idea on these extreme cases may be helpful for policy makers for different values of the effective reproduction number.

Reviewer #2:

Remarks to the Author:

I previously reviewed this manuscript for another journal, and the authors have done a good job in their reviewer response. I do not fully agree with the conceptual argument about the branching process model, but accept that this is the best available at present, and the ideas are important and should be published. The extra analyses and caveats also improve the analysis. I have no further comments on the modelling details.

I am still concerned with the tone of the manuscript, and especially that a particular policy is being 'sold' rather than a set of modelling results presented in an unbiased way. In my opinion, the title, abstract and subheadings all oversell a set of results which, while novel, are nonetheless derived from a highly simplified model which has limited scope.

As an example, the subheading "Bidirectional manual tracing with an expanded tracing window can more than double efficacy" implies that similar quantitative gains would be made in real-world scenarios, and I don't think that the evidence is there to back this up. And the 275% doesn't really reflect a difference between scenarios that would exist in the real world (because of the idealised way in which forward vs reverse tracing is modeled).

Personally, I think that the manuscript is novel and interesting enough, without the need for grand claims. I suggest that the title and subheadings are toned down, and that the introduction and abstract are caveated in a similar way to the discussion. However, I appreciate that these are ultimately subjective opinions, and perhaps something for discussion between the authors and editor.

Final revisions for Nature Communications manuscript NCOMMS-20-41043-T

Reviewer #1 (Remarks to the Author):

I am happy with the answers of the authors. My only point remains that the costs involved with backward tracing (a larger time window) are not quantified.

In the answers to the reviewers, the authors state: "There is also no simple relationship between the number of cases discovered and the number of individuals tested/quarantined, as the latter depends on the network structure of the population, the number and frequency of recurring contacts, and the manner (e.g. behavioural vs physiological) in which superspreaders differ from the rest of the population. As a result, it is difficult to quantitatively assess the costs of bidirectional tracing without a model that explicitly distinguishes between contacts and cases."

I fully agree with this statement, and it would be good that this is mentioned in the main text as well, which only states: "Our model, which takes no account of uninfected individuals or repeated contacts, is ill equipped to quantify this aspect of the policy question."

We thank the reviewer for the suggestion, and have updated the language of the manuscript accordingly.

I do not think it is essential for publication, but despite the fact that the model is ill-equipped, it may be possible to obtain some estimates of the costs associated with different time windows, e.g.,

1) what is the ratio of detected patients per time unit with the different strategies (this ignores the fact that many contacts who did not acquire the disease got screened).

2) Each individual has the same very high number of contacts per unit of time and there is random mixing in the population. Then the number of patients screened should be proportional to the number of patients for which the contacts are screened multiplies by the length of the time window.

My guess is that the true effort will be somewhere between these two cases, but having an idea on these extreme cases may be helpful for policy makers for different values of the effective reproduction number.

We agree that this would be desirable. We have added a sentence discussing this upper bound scenario to the discussion section. We hope to investigate this in more detail in future work.

Reviewer #2 (Remarks to the Author):

I previously reviewed this manuscript for another journal, and the authors have done a good job in their reviewer response. I do not fully agree with the conceptual argument about the branching process model, but accept that this is the best available at present, and the ideas are

important and should be published. The extra analyses and caveats also improve the analysis. I have no further comments on the modelling details.

I am still concerned with the tone of the manuscript, and especially that a particular policy is being 'sold' rather than a set of modelling results presented in an unbiased way. In my opinion, the title, abstract and subheadings all oversell a set of results which, while novel, are nonetheless derived from a highly simplified model which has limited scope.

As an example, the subheading "Bidirectional manual tracing with an expanded tracing window can more than double efficacy" implies that similar quantitative gains would be made in real-world scenarios, and I don't think that the evidence is there to back this up. And the 275% doesn't really reflect a difference between scenarios that would exist in the real world (because of the idealised way in which forward vs reverse tracing is modeled).

Personally, I think that the manuscript is novel and interesting enough, without the need for grand claims. I suggest that the title and subheadings are toned down, and that the introduction and abstract are caveated in a similar way to the discussion. However, I appreciate that these are ultimately subjective opinions, and perhaps something for discussion between the authors and editor.

We thank the reviewer for the forthright advice. We are sensitive to the trade-off between communicating our results clearly and making sure the necessary caveats are included. Our model is indeed simplified, but we have done our best to acknowledge its limitations throughout the manuscript, especially in the discussion.

Nevertheless, we have made several changes to tone down the text at key points, as discussed in more detail below.

Regarding the particular section of the results mentioned by the reviewer, we have edited the text to clarify that we are not predicting a 275% improvement in R_{eff} reduction. As members of Deirdre Hollingsworth's group at Oxford (among others) have pointed out, while current tracing programs are not explicitly aiming to carry out bidirectional tracing, some limited degree of reverse tracing is probably happening by accident. As such, real tracing programs probably fall somewhere between our 2-day forward-only and 2-day bidirectional scenarios, such that increasing the tracing window to 6 days pre-onset would (given our other assumptions) increase the reduction in R_{eff} by somewhere between 85 and 275%.

We have added a sentence in the manuscript to make this clear. However, since most of that range is over 100%, we stand by our claim that 6-day bidirectional tracing could "more than double efficacy". Nevertheless, we have toned down the subheading slightly and added additional caveats to the last paragraph of the section.

Regarding the rest of the results, it is admittedly difficult to communicate nuance clearly in short subheadings, and we are sensitive to the trade-off between communicating our results clearly

and making sure the necessary caveats are included. We have made some changes to the subheaders to tone them down: for example, replacing “benefits” with “predicted benefits” and “is robust” with “is more robust”. We believe the text of these subsections is, for the most part, sufficiently caveated already.

Regarding the rest of the text, we stand by our claims in the final sentence of the introduction, which is fairly mild. For the title, we have replaced “dramatically improves” with “could dramatically improve”, which we believe expresses greater uncertainty about translating the results of the model into reality while reflecting the genuinely dramatic effect we observe. Overall, we have done our best to add appropriate caveats while maximizing the likelihood that our model spurs real-world changes. We’re happy to discuss this balance with the editor.